# Majorana Excitons in a Kitaev Chain of Semiconductor Quantum Dots in a Nanowire

**DOI:** 10.3390/nano13162293

**Published:** 2023-08-09

**Authors:** Mahan Mohseni, Hassan Allami, Daniel Miravet, David J. Gayowsky, Marek Korkusinski, Pawel Hawrylak

**Affiliations:** 1Department of Physics, University of Ottawa, Ottawa, ON K1N 6N5, Canada; mghaf073@uottawa.ca (M.M.); dmiravet@gmail.com (D.M.); sgayo008@uottawa.ca (D.J.G.); marek.korkusinski@nrc-cnrc.gc.ca (M.K.); 2Security and Disruptive Technologies, National Research Council, Ottawa, ON K1A 0R6, Canada

**Keywords:** semiconductor quantum dots, Majorana and bond Fermions, Majorana zero mode, topological superconductor, excitons, electron-electron interactions, light–matter interaction

## Abstract

We present here a theory of Majorana excitons, photo-excited conduction electron-valence band hole pairs, interacting with Majorana Fermions in a Kitaev chain of semiconductor quantum dots embedded in a nanowire. Using analytical tools and exact diagonalization methods, we identify the presence of Majorana zero modes in the nanowire absorption spectra.

## 1. Introduction

There is currently interest in realizing synthetic topological quantum matter with topologically protected quasiparticles at its edges [1,2,3], with potential application in topological quantum computation [4,5,6,7,8,9]. Haldane fractional spin quasiparticles in a spin-one chain and Majorana Fermions in topological superconductors are good examples [9,10,11]. To realize Majorana Fermions, Kitaev proposed [11,12] a chain of quantum dots on a p-wave superconductor that carries such non-local zero energy Majorana Fermions localized on its two ends, the Majorana zero modes (MZMs). Since then there have been numerous proposals to realize the Kitaev chain [13,14,15,16,17,18,19]. In all cases, experimental confirmation of the presence of the MZMs has proven to be a non-trivial and challenging task [20,21,22,23,24,25,26,27,28].

Recent progress in semiconductor quantum dots in nanowires [29,30,31,32,33,34,35,36] opens the possibility of realizing Kitaev chains and optical detection of their Majorana zero modes. In this work, we consider such an array of InAsP quantum dots embedded in an InP nanowire as the material system [29,30,31,32,33,34,35,36] for realization of MZM and study its signature in light–matter interaction. As the schematic in Figure 1a shows, we combine a semiconductor nanowire with a p-wave superconductor [37,38,39,40,41,42]. The p-wave pairing in this system is introduced by the proximity effect among electrons that are spin-polarized by an external magnetic field, making sure that Cooper pairs can only form between electrons in the conduction band (CB) of adjacent dots. We will show that one can tune the system parameters into a topological regime, where two MZMs appear at the two ends of the chain. With semiconductor quantum dots, light can generate a hole in the valence band (VB) and an electron in the conduction band. The electron adds to an existing gas of Majorana Fermions while the hole then interacts with all the quasiparticles of the Kitaev chain, including MZMs, to form composite objects similar to excitons and trions in the Fermi Edge Singularity problem [43,44,45,46]. This leads to a structure in the absorption spectrum of the chain as a function of photon energy. Here we present a theory for the signatures of the MZMs in the optical spectra of the semiconductor nanowire.

After describing the model in Section 2, in Section 3 we introduce the exact diagonalization (ED) method and introduce Majorana and bond Fermion representation of the Kitaev Hamiltonian. Next, in Section 4 we describe exciton–Majorana Fermion complexes and predict the absorption spectrum. We focus discussion on the optical signature of the MZM in the absorption spectrum. Finally, in Section 5 we conclude by summarising our results and discuss potential experiments detecting Majorana Fermions in a semiconductor Kitaev chain.

## 2. Kitaev Chain in a Semiconductor Nanowire

Figure 1a shows a schematic representation of the Kitaev chain we are considering. It consists of a hexagonal InP nanowire with an array of embedded InAsP quantum dots in the proximity of a p-wave superconductor [37,38,39,40,41,42], in the presence of an applied external magnetic field. Such arrays have been extensively investigated [29,30,31,32,33,34,35,36], including their excitonic complexes [29,36]. The current advanced fabrication techniques allow for controlling various aspects of this design. The challenging part, which requires more experimental efforts, is the induction of p-wave superconductivity in the chain. As Figure 1b shows, in our model we include the lowest conduction spin level of each dot and the highest spin valence band level, which are effectively both spin-polarized due to the external magnetic field. Consequently, in the presence of superconductivity, only the electrons from the adjacent conduction levels can pair up, as there is only one conduction level available in each dot. The magnetic field should be large enough to cause a Zeeman splitting larger than hopping and pairing, but smaller than level separation in each dot. A magnetic field in any direction can provide the desired Zeeman splitting, but by choosing it along the wire we avoid unnecessary complications related to Landau quantization and level repulsion in neighboring dots. As the incident photon in Figure 1 represents, the spectroscopy of the Kitaev chain is assumed to be carried out with photons incident along the wire growth. This is dictated by embedding nanowire in an InP shell, which confines photons and increases light–matter coupling.

The Kitaev Hamiltonian, $H_{e}$, in Equation ([Disp-formula FD1a-nanomaterials-13-02293]) describes the hopping and pairing of electrons in conduction band levels. The chemical potential is tuned to bring the chain to near half filling in the absence of superconductivity. Therefore, in the equilibrium the valence levels are full and the system is described by the Kitaev Hamiltonian. However, when a photon with energy close to the band gap of InAsP illuminates the dots, it generates a hole in VB and an electron in CB. Adding an electron to the superconducting ground state excites its quasiparticles. The hole then forms a bound state with the quasiparticles of the electronic system that is in a collective superconducting state. These possible bound states generate peaks in the absorption spectrum of the system, among which there is the signature of MZM, as we shall show below.

The hole is described by a simple tight binding Hamiltonian, $H_{h}$, in Equation (1b). We also consider electron–hole interaction, $H_{int}$, in Equation (1c), which is the strongest when both conduction electron and valence band hole are on the same quantum dot. Hence, we write the full Hamiltonian as
$$\begin{array}{cc} \; & H=H_{e}+H_{h}+H_{int}, \end{array}$$
(1a)$$\begin{array}{cc} \; & H_{e}=t\sum_{i=1}^{N-1}c_{i+1}^{†}c_{i}+h.c.+\Delta \sum_{i=1}^{N-1}c_{i+1}^{†}c_{i}^{†}+h.c.-\mu \sum_{i=1}^{N}c_{i}^{†}c_{i}, \end{array}$$
(1b)$$\begin{array}{cc} \; & H_{h}=-\tau \sum_{i=1}^{N-1}h_{i+1}^{†}h_{i}+h.c.+\eta \sum_{i=1}^{N}h_{i}^{†}h_{i}, \end{array}$$
(1c)$$\begin{array}{cc} \; & H_{int}=-V\sum_{i=1}^{N}n_{i}^{e}n_{i}^{h}, \end{array}$$
where $c_{i}^{†}\left(h_{i}^{†}\right)$ is the normal Fermionic creation operator of an electron(hole) in dot *i*, t(τ) is hopping between adjacent conduction (valence) levels, Δ is the pairing energy between adjacent conduction levels, μ is the chemical potential measured from the conduction energy level, and η is the CB to VB energy gap in each dot. In the interaction term Equation (1c), *V* is the Coulomb attraction energy between electrons and holes, where we also introduced $n_{i}^{e}=c_{i}^{†}c_{i}\left(n_{i}^{h}=h_{i}^{†}h_{i}\right)$, the electron (hole) number operator in dot *i*. Figure 1b schematically shows different terms of Equation (1) between two adjacent dots.

Throughout the paper, we are going to express energies in the units of |t|, which can be tuned in the range of μeV to meV [30,47]. The electron–hole attraction, *V*, can be tuned independently and is much larger than |t| in typical designs [30,36], while hole hopping energy, τ, is expected to be much smaller than |t| as VB effective mass is much larger than CB effective mass [48]. The onsite gap, η, is of the order of the bandgap of InAs and is in the eV range [48]. The pairing Δ is also expected to be in the µeV–meV range [37,38,39,40,41,42], and since *t* is more controllable, one would need to tune *t* accordingly to bring the system into topological regime.

Next, before describing the absorption experiment, we start with a brief discussion of Kitaev Hamiltonian.

## 3. Majorana and Bond Fermions in Kitaev Hamiltonian

The Kitaev Hamiltonian, $H_{e}$, in Equation ([Disp-formula FD1a-nanomaterials-13-02293]), originally introduced in Ref. [11], supports two MZMs localized on the two ends of the chain, when the Hamiltonian is in topological regime. For a finite chain, the topological region is centered on parameters Δ=t and μ=0 [11], which is our focus throughout this work. Here, after describing the exact diagonalization (ED) method for normal Fermions, following Kitaev [11], we show how using Majorana Fermions reveals the usefulness of a new set of Fermions we refer to as *bond Fermions*. Next, after matching energy spectra obtained by ED in both normal and bond Fermion bases, we shall use the bond Fermion basis for the rest of the paper.

### 3.1. Exact Diagonalization in Normal Fermion Basis

We start off by introducing the exact diagonalization method (ED) for finding the energy spectrum of the Kitaev Hamiltonian. In ED, we span the Hilbert space of the system by configuration basis [49]. For our electronic system being made of *N* spinless orbitals, there are N0+N1+…+NN−1+NN=2N possible configurations, which we construct as
(2)$$|\alpha_{1}\ldots \alpha_{N}\rangle =\prod_{i=1}^{N}{\left(c_{i}^{†}\right)}^{\alpha_{i}}|0\rangle ,$$
where $|0\rangle$ is the vacuum of electrons, $\alpha_{i}=1\;\mathrm{or}\;0$, which corresponds to having (1) or not having (0) an electron in orbital *i*.

For a given number of electrons, *M*, we generate electron configurations, $p_{M}$. However, as Kitaev Hamiltonian, being a Hamiltonian for a superconductor, does not conserve the particle number, its eigenstates are coherent linear combinations of electronic configurations with different electron numbers as
(3)$$|\psi^{\nu }\rangle =\sum_{M,p_{M}}C_{M,p_{M}}^{\nu }|M,p_{M}\rangle ,$$
where we are populating *N* sites with M=0,1,…,N electrons. To solve for coefficients $C_{M,p_{M}}^{\nu }$, we apply the Hamiltonian on this state, and by using the orthogonality of the configurations we obtain the eigenvalue equation
(4)$$\sum_{p_{M},M}\langle q_{M^{\prime }},M^{\prime }|H_{e}|p_{M},M\rangle C_{M,p_{M}}^{\nu }=E^{\nu }C_{M^{\prime },q_{M^{\prime }}}^{\nu }.$$

However, since the Kitaev Hamiltonian, $H_{e}$, in Equation ([Disp-formula FD1a-nanomaterials-13-02293]) only changes particle number in pairs, the matrix element $\langle q_{M^{\prime }},M^{\prime }|H_{e}|p_{M},M\rangle$ is non-zero only if *M* and $M^{\prime }$ have the same parity, i.e., if they are both even or odd. This parity symmetry allows us to break the Hilbert space into two decoupled subspaces of even and odd configurations. In Appendix A we explicitly show the configurations and the Hamiltonian matrix $\langle q_{M^{\prime }},M^{\prime }|H_{e}|p_{M},M\rangle$ in each of these subspaces, for the case of N=3.

### 3.2. Bond Fermions

We now express the Kitaev Hamiltonian in Equation ([Disp-formula FD1a-nanomaterials-13-02293]) in terms of Majorana and bond Fermions. First, as schematically shown in Figure 2, we write each electron operator, *c* and $c^{+}$, in terms of two Majorana Fermion operators, $\gamma_{1}$ and $\gamma_{2}$, as
(5)$$\begin{array}{ccc} \; & c_{j}=\frac{1}{2}\left(\gamma_{j,1}+i\gamma_{j,2}\right),\; & c_{j}^{†}=\frac{1}{2}\left(\gamma_{j,1}-i\gamma_{j,2}\right), \end{array}$$
where the γ’s are Majorana Fermion operators. Majorana Fermions satisfy a slightly different anti-commutation relation than ordinary Fermions, $\left\{\gamma_{i,\alpha },\gamma_{j,\beta }\right\}=2\delta_{ij}\delta_{\alpha \beta }$.

Using Equation (Equation 5) and Majorana anti-commutation relations, the Hamiltonian, $H_{e}$, can be written in terms of Majorana Fermions as
(6)$$H_{e}=\frac{i}{2}\left(\left(t+\Delta \right)\sum_{j=1}^{N-1}\gamma_{j,1}\gamma_{j+1,2}+\left(t-\Delta \right)\sum_{j=1}^{N-1}\gamma_{j+1,1}\gamma_{j,2}-\mu \sum_{j=1}^{N}\left(\gamma_{j,1}\gamma_{j,2}-i\right)\right).$$

The form in Equation (Equation 6) shows the pairing between Majoranas of different types in adjacent sites. While the Hamiltonian is not diagonal in terms of Majoranas, the new pairing scheme suggests the introduction of a new set of auxiliary Fermions to diagonalize the Hamiltonian, as generally for two Majorana of different types we have $2c^{†}c-1=i\gamma_{2}\gamma_{1}$. Hence, following Kitaev [11], as shown in Figure 2, we define a new set of Fermionic operators, bond Fermions, which are made of two Majoranas of different types from adjacent sites as
(7a)$$\begin{array}{cc} \; & a_{j}=\frac{1}{2}\left(\gamma_{j,1}+i\gamma_{j+1,2}\right)=\frac{1}{2}\left(c_{j}^{†}+c_{j}+c_{j+1}-c_{j+1}^{†}\right), \end{array}$$
(7b)$$\begin{array}{cc} \; & a_{N}=\frac{1}{2}\left(\gamma_{N,1}+i\gamma_{1,2}\right)=\frac{1}{2}\left(c_{N}^{†}+c_{N}+c_{1}-c_{1}^{†}\right), \end{array}$$
where we also defined $a_{N}$, to which we refer as the zero mode, out of the two unpaired Majoranas at the two ends of the chain, as shown in Figure 2. Then, the Hamiltonian in terms of bond Fermion operators is
(8)$$\begin{array}{cc} H_{e}=\frac{1}{2} & \left(\left(t+\Delta \right)\sum_{j=1}^{N-1}\left(2a_{j}^{†}a_{j}-1\right)+\left(t-\Delta \right)\sum_{j=1}^{N-1}\left(a_{j+1}^{†}a_{j-1}+a_{j+1}a_{j-1}+h.c.\right)\right.\\ \; & \left.-\mu \sum_{j=1}^{N}\left(1+\left(a_{j}^{†}a_{j-1}+a_{j}a_{j-1}+h.c.\right)\right)\right), \end{array}$$
where in the second and the third sum one should identify $a_{0}\equiv a_{N}$. Note that in the topological regime, when t=Δ and μ=0, the bond Fermions diagonalize the Hamiltonian in Equation (Equation 8) and reduce it to
(9)$$H_{e}=t\sum_{j=1}^{N-1}\left(2a_{j}^{†}a_{j}-1\right),$$
which implies a set of N−1 quasiparticles with energy 2t, and one non-local quasiparticle, $a_{N}$, with zero energy, hence the name zero mode. In this case, since bond Fermions are the quasiparticles of Kitaev Hamiltonian, their configurations are the eigenstates of the system.

In this spirit, we also use bond Fermion configurations for exact diagonalization of Kitaev Hamiltonian. In the same fashion as in Equation (Equation 2), we define bond Fermion configurations as
(10)$$|\bar{\alpha_{1}\ldots \alpha_{N}}\rangle =\prod_{i=1}^{N}{\left(a_{i}^{†}\right)}^{\alpha_{i}}|0_{a}\rangle ,$$
where $|0_{a}\rangle$ is the vacuum of bond Fermions, and we used the *overline* to distinguish these configurations from the normal Fermion configurations. Next, an equation similar to the equation in Equation (Equation 3) can be written for the eigenstates of the Hamiltonian in terms of bond Fermion configurations, where now $|M,p_{M}\rangle$ would represent the $p_{M}$ configuration of having *M* bond Fermions. Similar to the case of normal Fermions, since $H_{e}$ in Equation (Equation 8) also conserves the parity of bond Fermion numbers, we can split the Hilbert space into even and odd subspaces. In Appendix A, we explicitly show the bond Fermion configurations and the Hamiltonian matrix of Equation (Equation 8) in each of these subspaces, for the case of N=3.

### 3.3. Energy Spectrum

To demonstrate the usefulness of bond Fermion basis, we now describe the energy spectrum of a chain of N=3 quantum dots, obtained on both the normal and bond Fermion basis. Figure 3 shows the energy spectrum for the case of Δ=t<0. Throughout this work, we consider t<0, as it is the case for conduction bands hopping integrals. As we mentioned above, in this case, the configurations of bond Fermions are also the eigenstates of the system. Being in a topological regime, with these parameters the system has a doubly degenerate ground state, one in the odd subspace, $|\mathrm{GS}\rangle =|\bar{111}\rangle$, with all bond Fermions, and the other in the even subspace, $|\bar{\mathrm{GS}}\rangle =|\bar{110}\rangle$, which is missing the zero energy bond Fermion, $a_{N}\equiv a_{3}$. Next, we have the singly excited states, missing one non-zero bond Fermion, with excitation energy 2|t|, from which we have two in each subspace: $|a_{1}\rangle =|\bar{011}\rangle$ and $|a_{2}\rangle =|\bar{101}\rangle$ in the even subspace and $|\bar{a_{1}}\rangle =|\bar{010}\rangle$ and $|\bar{a_{2}}\rangle =|\bar{100}\rangle$ in the odd subspace. Finally, in each subspace, there is one doubly excited state, missing two non-zero bond Fermion with excitation energy 4|t|, $|\bar{a_{1}a_{2}}\rangle =|\bar{000}\rangle$ in the even subspace and $|a_{1}a_{2}\rangle =|\bar{001}\rangle$ in the odd subspace. Table 1 summarizes the description of the spectrum in terms of bond Fermions.

## 4. Kitaev Chain and a Light Induced Valence Hole

Absorption of a photon injects an electron–hole pair into the system. Therefore, the relevant optically excited states live in the subspace of all configurations with one hole. Here, after studying the energy spectrum of the full Hamiltonian in Equation (1) with one hole in the configuration space of bond Fermions, we discuss the absorption spectrum of the chain and the optical signature of the MZM.

### 4.1. Exact Diagonalization of Electron–Hole System

Having demonstrated the benefit of the bond Fermion basis, we now study the Hamiltonian with one hole in the configuration basis of bond Fermions and one hole. Using $|M,p_{M};m\rangle$ to refer to *M* bond Fermions being in their $p_{M}$ configuration, and the hole being at site *m*, we can find the spectrum by solving an equation similar to Equation (Equation 4), but considering the full Hamiltonian, *H*, in Equation (1) rather than $H_{e}$, and in the configuration basis of bond Fermions and one hole.

For instance, for N=3 dots, following the convention we introduced in Section 3.3 and Table 1, we can list these configurations as Table 2.

In this subspace, the hole Hamiltonian, $H_{h}$, in Equation (1b) amounts to a constant, η, and mixes states with the same electronic configurations and different locations of the hole by hopping matrix element τ. Therefore, with the ordering in Table 2, the full Hamiltonian with one hole for the example of N=3 dots has the following structure:(11)$$H=\left[\begin{array}{ccc} H_{1} & -\tau & 0\\ -\tau & H_{2} & -\tau \\ 0 & -\tau & H_{3} \end{array}\right]+\eta ,$$
where each block is a 4×4 matrix, τ is the identity matrix times τ, and the diagonal blocks are given by the matrix elements of $H_{e}+H_{int}$ in Equation (1) over the configurations in Table 2. The interaction term $H_{int}$ in Equation (1c) for each of the diagonal blocks, $H_{j}$, is $-Vn_{j}^{e}$, and it mixes up different bond Fermion configurations as we have
(12a)$$\begin{array}{ccc} n_{j}^{e} & =\frac{1}{2}+\frac{1}{2}\left(a_{j-1}^{†}a_{j}+a_{j-1}^{†}a_{j}^{†}+h.c.\right),\; & 1<j\leq N, \end{array}$$
(12b)$$\begin{array}{ccc} n_{1}^{e} & =\frac{1}{2}+\frac{1}{2}\left(a_{N}^{†}a_{1}+a_{N}^{†}a_{1}^{†}+h.c.\right),\; & j=1, \end{array}$$
which implies that when the hole is not at the two ends of the chain then the interaction mixes up two non-zero bond Fermions, and when it is at one of the two ends, the interaction mixes the zero mode with one of the non-zero ones. For instance, for the operators $n_{1}^{e}$ and $n_{2}^{e}$ in the even configuration basis in Table 2 we have
(13)$$\begin{array}{cc} n_{1}^{e}=\frac{1}{2}\left[\begin{array}{cccc} 1 & -1 & 0 & 0\\ -1 & 1 & 0 & 0\\ 0 & 0 & 1 & -1\\ 0 & 0 & -1 & 1 \end{array}\right],\; & n_{2}^{e}=\frac{1}{2}\left[\begin{array}{cccc} 1 & 0 & 0 & 1\\ 0 & 1 & -1 & 0\\ 0 & -1 & 1 & 0\\ 1 & 0 & 0 & 1 \end{array}\right], \end{array}$$
while $n_{3}^{e}$, in a similar fashion to $n_{1}^{e}$, mixes $|\bar{\mathrm{GS}}\rangle$ with $|a_{2}\rangle$ and $|a_{1}\rangle$ with $|\bar{a_{1}a_{2}}\rangle$.

### 4.2. Energy Spectrum of the Electron–Hole System

Figure 4 shows the energy spectrum of a chain of length N=3 dots in the even subspace and for Δ=t and μ=0, as the electron–hole interaction *V* increases, for a localized hole (τ=0) on the left panel and for a mobile hole with τ=0.3|t| on the right panel. Both cases show branching into two groups, pertaining to bonding and antibonding pairs of states, mixed by the interaction *V*.

The case of a localized hole allows us to understand the spectrum better. There are four electronic states associated with the valence hole being at each dot, and since the two end dots are geometrically the same, the spectrum always shows four pairs of doubly degenerate states. As we show in Appendix B, and it can also be seen from Equation (Equation 13), for the two end dots, two of these four states are mixtures of $|\bar{\mathrm{GS}};1\left(3\right)\rangle$ and $|a_{1(2)};1\left(3\right)\rangle$ that give us visible peaks at $E_{\pm }$, as described in Section 4.3.1, and also indicated on the plot. The two other pairs of degenerate levels are mixtures of $|a_{2(1)};1\left(3\right)\rangle$ and $|\bar{a_{1}a_{2}};1\left(3\right)\rangle$, which do not get excited by absorbing a photon, as the polarization operator $c_{1(3)}^{†}h_{1(3)}^{†}$ does not couple them to the ground state (see Appendix B). For the middle dot, as can be seen from Equation (Equation 13), one pair of states are mixture of $|a_{1};2\rangle$ and $|a_{2};2\rangle$, where only the bonded state gets excited by absorbing a photon (see Appendix B), resulting in the peak $E_{0}$; this is also described in Section 4.3.1 and shown on the plot. Finally, the last pair of states, which also do not get excited by absorbing a photon, are a mixtures of $|\bar{\mathrm{GS}};2\rangle$ and $|\bar{a_{1}a_{2}};1\left(3\right)\rangle$.

As can be seen in the right panel of Figure 4, for a mobile hole, when τ≠0, we still have one pair of doubly degenerate states as a result of the chain’s spatial symmetry. At V=0 for the τ=0 case, there is an extra triple degeneracy because of the non-dispersive nature of the localized hole band. However, for a mobile hole, it can be seen on the right panel of Figure 4 that at V=0 the degenerate levels split into sets of triples, corresponding to the three propagating modes of the hole band. More importantly, in this case, since the three dot subspaces are connected by hole hopping (see Equation (Equation 11)), the above described pairs of states mix up by τ, and the ones that are closer in energy mix more. As a result of this mixture, more peaks arise in the absorption spectrum, as we discuss in the next section.

### 4.3. Absorption Spectrum

As the schematic in Figure 1a shows, in the absorption experiment a photon probes the chain along the nanowire. InAsP dots have a significantly smaller bandgap than the InP bulk of the nanowire [48], which guaranties that the photon can only be absorbed by the dots. For calculating the absorption spectrum of the chain, we assume that the photon creates an electron–hole pair with uniform probability along the nanowire, and so define the polarization operator as
(14)$$P=\frac{1}{\sqrt{N}}\sum_{i=1}^{N}c_{i}^{†}h_{i}^{†}=\frac{1}{\sqrt{N}}\sum_{i=1}^{N}P_{i},$$
where we also introduced the local electron–hole pair operator $P_{i}=c_{i}^{†}h_{i}^{†}$. The strength of the polarization operator is determined by the dipole matrix element *d*, as well as the light polarization of the photon. The direction of the magnetic field with respect to the incident wave determines which helicity of photon is absorbed better. By choosing the form in Equation (Equation 14), we are factoring out the helicity of the incident photon and normalizing the result to ${\left|d\right|}^{2}$, as these multiplicative factors do not change the absorption profile.

We are assuming that one can also set up the system to create the electron–hole pair on a chosen specific dot, *i* [35,50,51,52], i.e., acting with the operator $P_{i}$ on the chain, rather than *P*. As we discussed, having access to such a spatially resolved spectrum is important in detecting the optical signature of the MZM.

The polarization operator $P_{\left(i\right)}$—we use this notation to simultaneously refer to *P* and $P_{i}$—takes the ground state of the system to an excited state with one hole and different electron parity. Since the ground state can be degenerate, as it is when Δ=t and μ=0, the absorption spectrum has an even and an odd part pertaining to each ground state
(15)$$\begin{array}{cc} A_{\left(i\right)}\left(E\right) & ={\left|\beta_{\mathrm{even}}\right|}^{2}\sum_{\phi_{\mathrm{odd}}}|\langle \phi_{\mathrm{odd}}{\left|P_{\left(i\right)}|{\mathrm{GS}}_{\mathrm{even}}\rangle \right|}^{2}\delta \left(E-E_{\phi_{\mathrm{odd}}}+E_{\mathrm{GS}}\right)\\ \; & +{\left|\beta_{\mathrm{odd}}\right|}^{2}\sum_{\phi_{\mathrm{even}}}|\langle \phi_{\mathrm{even}}{\left|P_{\left(i\right)}|{\mathrm{GS}}_{\mathrm{odd}}\rangle \right|}^{2}\delta \left(E-E_{\phi_{\mathrm{even}}}+E_{\mathrm{GS}}\right)\\ \; & ={\left|\beta_{\mathrm{even}}\right|}^{2}A_{\left(i\right)}^{\mathrm{even}}\left(E\right)+{\left|\beta_{\mathrm{odd}}\right|}^{2}A_{\left(i\right)}^{\mathrm{odd}}\left(E\right), \end{array}$$
where $|\phi_{\mathrm{even}(\mathrm{odd})}\rangle$ are the eigenstates of the one hole subspace and the corresponding electron parity, and we used the notation $A_{\left(i\right)}$ to simultaneously refer to the regular absorption spectrum *A*, and $A_{i}$ the spatially resolved absorption spectrum coming from dot *i*.

#### 4.3.1. Analytic Result for Localized Hole

If τ=0, the full Hamiltonian becomes block diagonal (see Equation (Equation 11)), i.e., the subspaces of having the hole in each of the dots decouple. Consequently, we have
(16)$$A\left(E\right)=\frac{1}{N}\sum_{i=1}^{N}A_{i}\left(E\right).$$

At the heart of the topological regime when Δ=t and μ=0 [11], as depicted graphically in Figure 5 and expressed in Equation (A6), an electron created at site *i* by $c_{i}^{†}$ decomposes into a superposition of creation and annihilation operators of the two bond Fermions on its two sides, $a_{i}^{\left(†\right)}$ and $a_{i-1}^{\left(†\right)}$. If the electron is created at one of the two ends, one of the bond Fermions is the zero mode $a_{N}$. On the other hand, when the hole is at site *i*, the interaction $-Vn_{i}^{e}n_{i}^{h}$ mixes the two bond Fermions, as shown in Figure 5 and expressed in Equations (Equation 26) and (Equation 30). As we show in Appendix B, combining these two mechanisms, one can find an analytic expression for the spatially resolved absorption spectrum for a chain of arbitrary length *N* if the hole is created on site *i* as
(17)$$A_{i}\left(E\right)=\frac{1}{2}\left\{\begin{array}{cc} \delta (E-E_{0}) & 1<i<N\\ A_{-}\delta \left(E-E_{-}\right)+A_{+}\delta \left(E-E_{+}\right) & i=1,N \end{array}\right.,$$
where
(18a)$$\begin{array}{cc} \; & E_{0}=\eta +2\left|t\right|-V, \end{array}$$
(18b)$$\begin{array}{cc} \; & E_{\pm }=\eta +\left|t\right|-\frac{V}{2}\pm \sqrt{t^{2}+{\left(\frac{V}{2}\right)}^{2}}, \end{array}$$
(18c)$$\begin{array}{cc} \; & A_{\pm }=\frac{1}{2}\left(1\mp \frac{V}{\sqrt{4t^{2}+V^{2}}}\right), \end{array}$$
and then the full absorption spectrum is given by the simple sum in Equation (Equation 16).

The bottom row of Figure 6 shows the results in Equations (17) and (18) for the case of N=3. The peak $E_{0}$ is only present in the middle, while the peaks $E_{\pm }$ are present at the two ends of the chain. As we show in Appendix B, the two peaks, $E_{\pm }$, have a mixture of zero mode in them, while $E_{0}$ is purely made of non-zero bond Fermions. At V=0, $E_{-}$ is purely made of zero mode while $E_{+}$ is purely made of non-zero bond Fermions. As we increase *V*, $E_{+}$ acquires more zero mode contribution, while $E_{-}$ mixes more with non-zero bond Fermions. At the same time, by increasing *V*, the peak at $E_{+}$ diminishes, as can be seen from Equation (18c). If not too weak, the $E_{+}$ peak is a better resolved optical signature for the MZM than $E_{-}$, as it is separated from the rest of the spectrum by *V*, and we expect to have V≫|t|. This presents an advantage over the scanning tunnelling microscopy approach for detecting MZM [21]. Moreover, if one can perform spatially resolved absorption spectroscopy on the chain, the presence of the zero mode can be determined by the presence of a visible peak at high energy near $E_{+}$ when probing the end dots, and its absence when probing other dots.

#### 4.3.2. Absorption for Mobile Hole

When the hole is mobile, there are *N* itinerant hole states with different energies. Therefore, one would expect *N* different transitions to each electronic state. More importantly, as can be seen in Equation (Equation 11), the hopping hole mixes up different subspaces of having the hole in different dots. As a result, more transitions are allowed, leading to the emergence of more peaks in the absorption spectrum.

In Figure 6, we compare the absorption spectrum of a mobile hole with τ=0.1|t| (top row), and the analytic result of Equation (Equation 17) for localized hole (bottom row), for the case of N=3. It is evident how more peaks are visible for the case of the mobile hole, while the major peaks are still close to the location of $E_{0}$ and $E_{\pm }$. Moreover, note how in the full spectrum $\bar{A}$ (top left) there is only one visible peak at high energy near $E_{+}$, and how the same peak is large in ${\bar{A}}_{1}$ (top middle) and faint in ${\bar{A}}_{2}$ (top right), pertaining to the localized nature of the MZM that $E_{+}$ carries. In plotting Figure 6, we used ${\bar{A}}_{\left(i\right)}=\frac{1}{2}\left(A_{\left(i\right)}^{\mathrm{even}}+A_{\left(i\right)}^{\mathrm{odd}}\right)$, as for a mobile hole the even and odd parts of the absorption spectrum are not the same. But since there is no preference between the two ground states, one would expect to observe an average of the two.

The same logic is valid for a chain of any length, as the analytic result in Equation (Equation 17) is for general *N*. The analytic results are particularly insightful as the computation complexity increases exponentially in the exact diagonalization method. In our system, after taking into account the parity symmetry of the Kitaev Hamiltonian, the size of the Hilbert space with one valence hole grows as $N2^{N-1}$. Figure 7 shows the absorption spectrum of a chain of length N=9. Here, we set V=10|t| while changing τ. When τ→0, we approach the idealized case of a localized hole, where the subspaces of having the hole on each dot are decoupled. Growing τ mixes up the modes of different dots. Consequently, the zero mode starts leaking out of the two ends of the chain. On the first panel of Figure 7, we can see that at high energy there is still only one visible peak near $E_{+}$, until about τ=0.3|t|, where a faint peak appears to the right of it. This makes $E_{+}$ a very robust signature for a relatively large range of hole hopping. Having access to spatially resolved spectrum, we can further confirm that the peak is indeed coming from the two ends. It can be seen from the third panel of Figure 7 that there is no visible high energy peak on the site next to the end dot (${\bar{A}}_{2}$) until around τ=0.1|t|. In contrast, we can observe in ${\bar{A}}_{2}$ that $E_{-}$, which also contains a large share of zero mode and is a stronger peak, starts leaking out of the end dot very quickly for small τ’s.

## 5. Conclusions

We present here a theory of Majorana excitons, photo-excited conduction electron-valence band hole pairs, interacting with Majorana Fermions in a Kitaev chain of semiconductor quantum dots embedded in a nanowire. We demonstrate how the excited states of the superconducting system can be represented by different configurations of bond Fermions, and using exact diagonalization techniques we compute the energy spectra of the system. We confirm the existence of nonlocal bond Fermion, a superposition of Majorana Fermions at the two ends of the chain, with zero energy. We introduce a valence band hole and describe its interaction with Majorana fermions. We predict interband absorption spectra and discuss the signature of Majorana zero modes in the absorption spectra. We demonstrate how a spatially resolved absorption spectrum can be used to confirm the localized character of the MZMs.

We hope this preliminary work motivates future theoretical and experimental work on hybrid nanowire semiconductor quantum dots/superconductor systems for the demonstration of Majorana Fermions.

## Figures and Tables

**Figure 1 nanomaterials-13-02293-f001:**
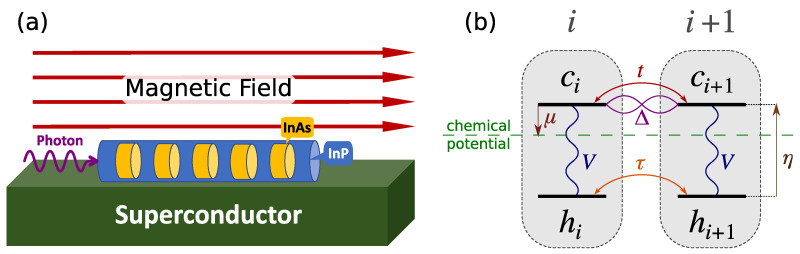
(**a**) Schematic of the system and the light absorption experiment. (**b**) Schematic of the Hamiltonian terms between two adjacent dots according to Equation (1), where conduction (valence) levels are labeled by $c_{i}\left(h_{i}\right)$ operators. The conduction level is the reference of energy, hence the downward arrow indicates negative μ. For TEM image and an atomistic description of the quantum dot nanowire system, see Refs. [30,34,36].

**Figure 2 nanomaterials-13-02293-f002:**
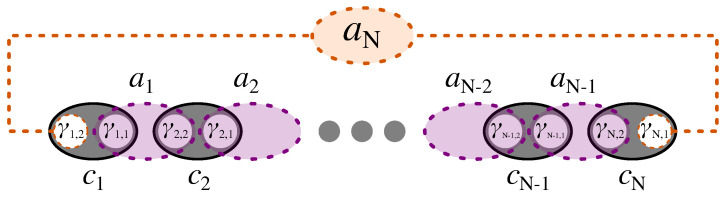
Schematic of Kitaev chain in the Majorana and bond representation, with non-zero bond Fermions in purple, and the nonlocal zero mode, $a_{N}$, living on the two ends of the chain.

**Figure 3 nanomaterials-13-02293-f003:**
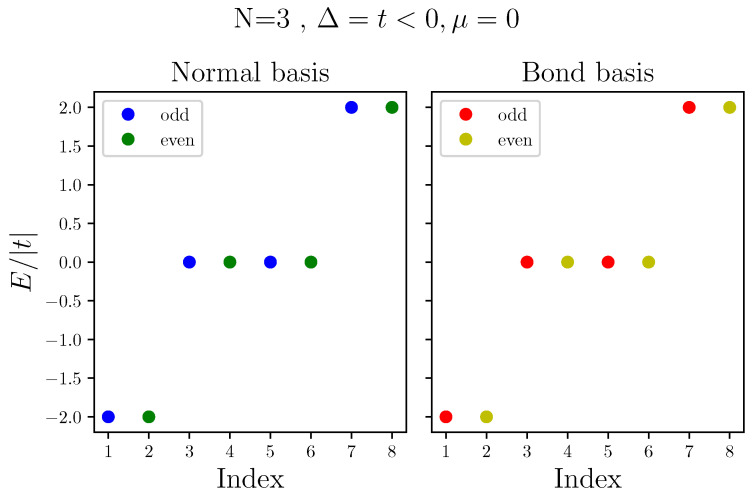
Energy spectra of Kitaev chain in normal (**left**) and bond (**right**) basis, where Δ=t<0 and μ=0. Energy is normalized to |t|.

**Figure 4 nanomaterials-13-02293-f004:**
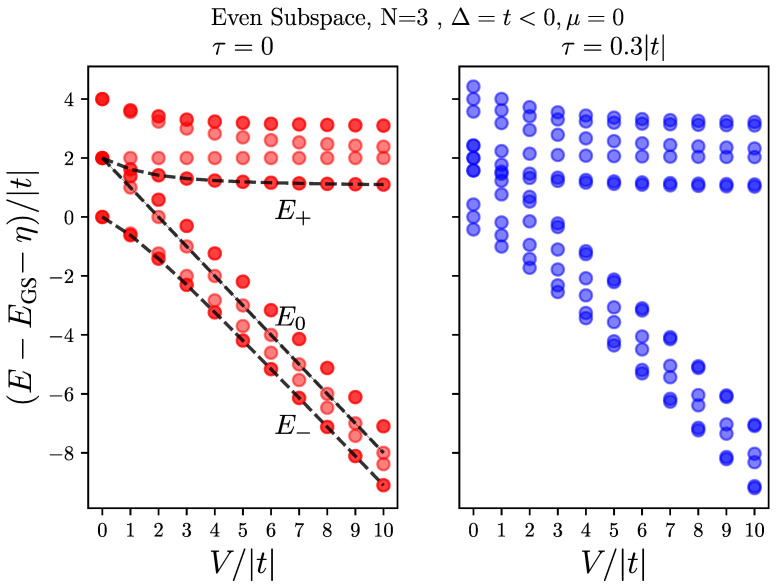
Energy spectra of the full Hamiltonian Equation (1) with one hole in the even subspace, as a function of electron–hole interaction, *V*, for N=3 dots, Δ=t, and μ=0. (**left**) for the case of localized hole, τ=0, (**right**) for a mobile hole with τ=0.3|t|. The overlap of transparent markers makes the degenerate levels look darker. The peak energies, $E_{0}$ and $E_{\pm }$, discussed in Section 4.3.1 are also shown according to Equation (18a,b).

**Figure 5 nanomaterials-13-02293-f005:**
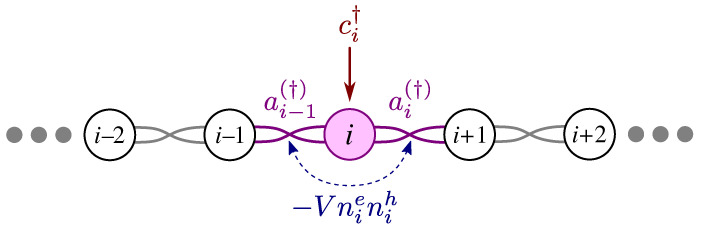
An electron created by $c_{i}^{†}$ is a superposition of creation and annihilation operators of two bond Fermions, $a_{i}^{\left(†\right)}$ and $a_{i-1}^{\left(†\right)}$, according to Equation (A6). The interaction $-Vn_{i}^{e}n_{i}^{h}$ mixes up the two bond Fermions according to Equation (Equation 26). Note that when *i* is one of the two ends, then one of the bond Fermions is the zero mode $a_{N}$ (see Appendix B for more details).

**Figure 6 nanomaterials-13-02293-f006:**
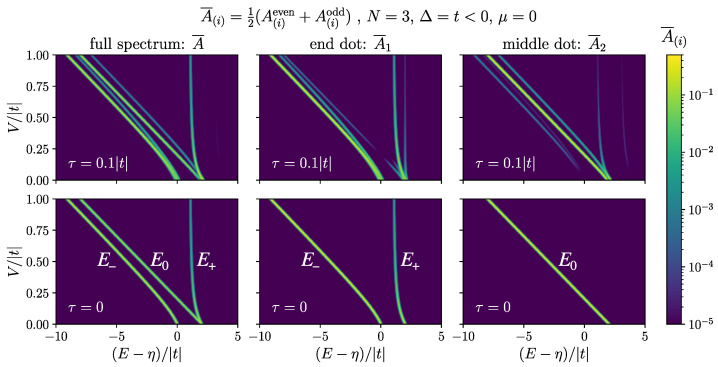
(**left**) The averaged absorption spectrum, $\bar{A}\left(E\right)$, and (**middle** and **right**) spatially resolved absorption, ${\bar{A}}_{i}\left(E\right)$, for Δ=t, μ=0, and for N=3 dots: (**top**) for a mobile hole with τ=0.1|t|, (**bottom**) for a localized hole, τ=0, according to the analytic results in Equations (17) and (18). The spectra are plotted against (E−η)/|t| while changing V/|t| on the y-axis. The bright curves show the location of the peaks as *V* changes, and the color scale shows their heights. Gaussian profile was used for the peaks with the width σ=0.025|t|. The maximum value of each peak shows the magnitude of the corresponding matrix element.

**Figure 7 nanomaterials-13-02293-f007:**
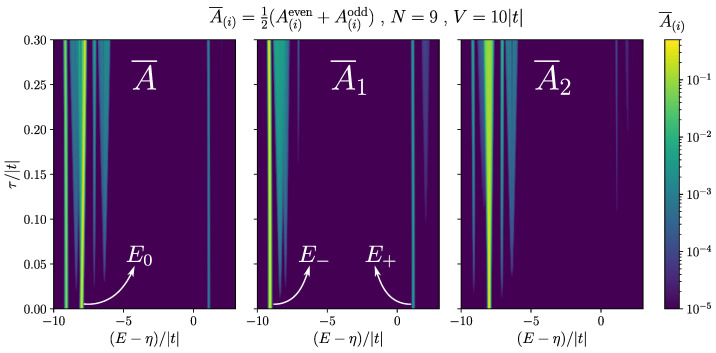
Absorption spectrum for a chain of length N=9, and for Δ=t, μ=0, V=10|t|, and changing τ. (**left**) The full averaged spectrum $\bar{A}$, (**middle**) the spatially resolved spectrum for the first dot ${\bar{A}}_{1}$, and (**right**) the spatially resolved spectrum for the second dot ${\bar{A}}_{2}$. The bright curves show the location of the peaks as τ changes, and the colorscale shows their heights. Gaussian profile was used for the peaks with the width σ=0.025|t|. The maximum value of each peak shows the magnitude of the corresponding matrix element.

**Table 1 nanomaterials-13-02293-t001:** Describing the spectra plotted in Figure 3. Configurations of bond Fermions are the eigenstates of Kitaev Hamiltonian when Δ=t and μ=0.

Index	1	2	3	4	5	6	7	8
Configuration	$|\bar{111}\rangle$	$|\bar{110}\rangle$	$|\bar{010}\rangle$	$|\bar{011}\rangle$	$|\bar{100}\rangle$	$|\bar{101}\rangle$	$|\bar{001}\rangle$	$|\bar{000}\rangle$
Label	$|\mathrm{GS}\rangle$	$|\bar{\mathrm{GS}}\rangle$	$|{\bar{a}}_{1}\rangle$	$|a_{1}\rangle$	$|{\bar{a}}_{2}\rangle$	$|a_{2}\rangle$	$|a_{1}a_{2}\rangle$	$|\bar{a_{1}a_{2}}\rangle$
Parity	odd	even	odd	even	odd	even	odd	even
Excitation								
Energy	0	0	2|t|	2|t|	2|t|	2|t|	4|t|	4|t|

**Table 2 nanomaterials-13-02293-t002:** Configurations of bond Fermions with one hole for N=3 dots.

Even				Odd			
$|\bar{\mathrm{GS}};1\rangle$	$|a_{1};1\rangle$	$|a_{2};1\rangle$	$|\bar{a_{1}a_{2}};1\rangle$	$|\mathrm{GS};1\rangle$	$|{\bar{a}}_{1};1\rangle$	$|{\bar{a}}_{2};1\rangle$	$|a_{1}a_{2};1\rangle$
$|\bar{\mathrm{GS}};2\rangle$	$|a_{1};2\rangle$	$|a_{2};2\rangle$	$|\bar{a_{1}a_{2}};2\rangle$	$|\mathrm{GS};2\rangle$	$|{\bar{a}}_{1};2\rangle$	$|{\bar{a}}_{2};2\rangle$	$|a_{1}a_{2};2\rangle$
$|\bar{\mathrm{GS}};3\rangle$	$|a_{1};3\rangle$	$|a_{2};3\rangle$	$|\bar{a_{1}a_{2}};3\rangle$	$|\mathrm{GS};3\rangle$	$|{\bar{a}}_{1};3\rangle$	$|{\bar{a}}_{2};3\rangle$	$|a_{1}a_{2};3\rangle$

## Data Availability

The codes used to produce data presented in this study are openly available in “Kitaev Exciton” at Ref. [53].

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
