# Peer review of "Majorana Excitons in a Kitaev Chain of Semiconductor Quantum Dots in a Nanowire"

_nanomaterials, 2023, doi:10.3390/nano13162293_

Round 1

Reviewer 1 Report

The work presents a theory study of Majorana exciton in a Kitaev chain of InAsP quantum dots embedded in InP nanowires. The theoretical framework developed the comprehensive understanding of Majorana excitons. Overall, the study provides a complete story about Majorana zero modes, but I believe that addressing the following points could further improve the understanding and application of the findings:

  1. What is the relationship between magnetic field strength and the realization of Majorana fermions in a Kitaev chain of InAsP quantum dots embedded in an InP nanowire? Specifically, what are the critical effects of magnetic field strength that are needed to observe Majorana zero modes in this system?
  2. How do the excitonic complexes of the embedded quantum dots affect the behavior of the Kitaev chain under the influence of an applied external magnetic field?
  3. The statement provided beneath Figure 2 discusses the nondiagonal nature of the Hamiltonian in Eq. (6) and its physical implications, but it may require further elaboration. Could the authors provide more detailed explanation about the nondiagonal nature of the Hamiltonian and its physical implications in the corresponding system?
  4. Could the authors please provide a more detailed explanation with physical reasoning, particularly regarding the mixed state and the interaction of incident photons, for the statement on lines 174-175? Additionally, would it be possible for the authors to modify this statement to provide more clarity.
  5. Considering the current demonstration, it appears that exact diagonalization techniques are effective for studying Majorana and bond fermions in one-dimensional chains and similar systems. However, what are the possible limitations of this approach when applied to larger arrays or more complex quantum systems?
  6. How feasible is the proposed system of InAsP quantum dots embedded in an InP nanowire for realizing Majorana zero modes compared to other proposed systems or materials?

Please check the grammer and sentence carefully.

Reviewer 2 Report

Majorana fermion has ever been reported in 2018, which should be a quite novel topic in quantum physics. The manuscript theoretically focused on Majorana Zero Modes in InAs/InP nanowires. I can recommend the publication after solving the following questions.

1. I cannot know the structures of nanowires clear, which seem to be a quantum-well-like heterostructure. The schematic illustrations of atomic structures of them may help a lot. The actual TEM images would be better to show whether it can be fabricated.

2. Figure 1 shows the magnetic field induced the transport of phonons. What will happen if the incident phonons were opposite?

3. What if the direction of the magnetic field was perpendicular to the phonon incident direction?

Reviewer 3 Report

The paper is aimed a the extension of Kitaev’s toy model of a quantum wire lying on the surface of three-dimensionalp-wave superconductor on the case of photo-excited conduction electron-valence  band hole pairs, interacting with Majorana Fermions. However there are several questions about substantiation  of the main provisions of the model and about the results, in particular:

1. The electron becomes one of the electrons in CB and decomposes into quasiparticles of

the superconducting state .  It  is known, that the superconducting state consists  of quasi-particles - Cooper pairs, which consist of two electrons. The pairs can decompose into electrons, but how one electron is decomposed into quasi-particles. It is also known that Cooper pairs are formed from the electrons of a Fermi surface. The cases when the pairs are formed from excited states are not known.    

2. Also the results obtained for superconductors are not clear from conclusion : We hope this preliminary work motivates future theoretical and experimental work on hybrid nanowire semiconductor quantum dots /superconductor systems for the demonstration of Majorana Fermions.

3. Authors Hamiltonian consists of 5 parameters. How can they estimate these parameters ?  Despite of large number of parameters of the model one important one is missing, namely dipolar part of Hint in (1c) is imaginary, since it corresponds to a radiative decay of the excited state.Also it is not clear why the same terms m and h for electrons and holes are denoted be different words chemical potential and energy.

4 The term bond Fermions looks to be incorrect. Perhaps fermionic bond or bound fermions ?

5. Being in topological regime, How author define the topological regime ? Why  the heart of topological regime when D = t and m = 0,

6. When the hole is mobile, there are N itinerant hole states with different energies. The itinerant hole in solid is described by a state with the definite wavector k and  energy.  Why the authors think that there are different energies ?

7. In figures 6 and 7  absorption spectra are presented as functions of (E-h)t . This parameters appear in eq(1) in expressions tc+i+1ci  and hh+ihi  . Perhaps there is some influence of these  two center electron-electron and  of one-center hole-hole interaction. However the main term defining the photoabsorbtion  is  <hi|d|ci>, where d is dipole matrix element of interaction with photon, is missing

Due to above mentioned reasons I can’t recommend this work for publication.

Round 2

Reviewer 3 Report

The author made some changes according to the first report and tha paper may be recommended for publising